# The Use of Ceftazidime–Avibactam in a Pediatric Intensive Care Unit—An Observational Prospective Study

**DOI:** 10.3390/antibiotics13111037

**Published:** 2024-11-03

**Authors:** Raquel García Romero, Elena Fresán-Ruiz, Carmina Guitart, Sara Bobillo-Perez, Iolanda Jordan

**Affiliations:** 1Pediatrics Department, Hospital Sant Joan de Déu, University of Barcelona, 08950 Barcelona, Spain; rgarciaro@sjd.es; 2Pediatric Intensive Care Unit, Hospital Sant Joan de Déu, University of Barcelona, 08950 Barcelona, Spain; elena.fresan@sjd.es (E.F.-R.); carmina.guitart@sjd.es (C.G.); sara.bobillo@sjd.es (S.B.-P.); 3Immunological and Respiratory Disorders in the Pediatric Critical Patient Research Group, Institut de Recerca Sant Joan de Déu, University of Barcelona, Hospital Sant Joan de Déu, 08950 Barcelona, Spain; 4Consorcio de Investigación Biomédica en Red de Epidemiología y Salud Pública (CIBERESP), 28029 Madrid, Spain

**Keywords:** antibiotics, ceftazidime–avibactam, MDR bacteria, CRE, carbapenemases, bacteremia, sepsis, PICU

## Abstract

**Background/objectives:** Infections caused by carbapenem-resistant Enterobacterales (CRE) are progressively increasing in Pediatric Intensive Care Units (PICUs). Its treatment is challenging due to the lack of pediatric trials. CRE infections are associated with significantly poor outcomes, but ceftazidime–avibactam (CAZ-AVI) has been reported to be successful in their treatment. This study aimed to describe the use and outcome of CAZ-AVI in a PICU. **Results:** Ten patients were included, with 12 episodes of clinical suspicion or confirmed multidrug-resistant (MDR) bacterial infections treated with CAZ-AVI for surgical prophylaxis, suspicion of sepsis, pneumonia, and surgical wound infection. Of these patients, 80% received empirical treatment because of previous MDR bacterial colonization, and 60% were administrated combination therapy with aztreonam for Metallo-β-Lactamases (MBL)strains. No bacteria were resistant to CAZ-AVI. The average duration of the treatment was 3 days when cultures turned negative and 7 days when MDR bacteria were isolated. **Methods:** This was an observational prospective study of children treated with CAZ-AVI in the PICU of a tertiary hospital in 2022. Epidemiological, clinical, microbiological, and outcome data were collected. **Conclusions:** The most frequent use of CAZ-AVI in our PICU was the short-term empirical treatment for patients with previous MDR bacterial colonization and clinical suspicion of bacteremia or sepsis. Furthermore, the combination of CAZ-AVI plus aztreonam could be more effective for CRE infections, especially type Ambler class B as MBL strains.

## 1. Introduction

Multidrug-resistant (MDR) bacterial infections include infections caused by carbapenem-resistant Enterobacterales (CRE), which are one of the urgent threats listed by the Centers for Disease Control and Prevention (CDC) with critical priority as determined by the World Health Organization (WHO) [1,2]. These infections are associated with significant morbidity and mortality due to a delay of appropriate antibiotic treatment and the need for an alternative treatment with less effectiveness and a worse safety profile [3,4]. In the last few decades, there has been an increase in CRE infections in hospitalized children, especially in complicated patients admitted to Pediatric Intensive Care Units (PICUs) which are linked to significantly poorer outcomes. Mortality rates in children with CRE infection have been reported as very variable, but neonates seem to be the group with the highest risk [5,6,7,8].

CRE are defined by the CDC as members of the Enterobacterales order resistant to at least one carbapenem antibiotic. CRE that produce carbapenemases, enzymes that break down carbapenems making them ineffective, are called carbapenemase-producing CRE (CP-CRE). Carbapenem resistance is developed by genes that hydrolyze the β-lactam ring of carbapenem antibiotics, by the production of extended-spectrum β-lactamases (ESBLs), like Ambler Class A serine *K.pneumoniae* carbapenemase (KPC), the Ambler class B carbapenemases that require zinc to be active as metallo-β-lactamases (MBLs) or Amp C β-lactamases combined with impaired membrane permeability [7]. The most common carbapenemase in the United States is the *Klebsiella pneumoniae* carbapenemase (KPC). In Spain, the most common carbapenemase is OXA 48, followed by metallo-β-lactamases (MBLs) and only 4% of invasive KPCs. Epidemiology in children is not exactly known, and pediatric data reported worldwide (including countries such as Spain, the United States or Italy) have mostly been related to the sporadic spread and outbreaks of CRE infections, except in those countries where CRE are highly endemic (e.g., India or Turkey). Common CRE isolated in children cultures include *Klebsiella pneumoniae* and *Escherichia coli*, among others [5,7,8,9,10,11].

Adequate treatment is one of the most relevant and potentially modifiable prognostic factors [12]. The treatment of CRE pediatric infections is challenging due to the lack of comparative studies and data based on studies conducted in adults. Many drug combinations have emerged over the last few years, making it necessary to update this topic in the pediatric field. The delay in performing pediatric trials is leading to off-label use in these patients. Ceftazidime–avibactam (CAZ-AVI) is a combination of third-generation cephalosporin (ceftazidime) and a novel synthetic β-lactamase inhibitor capable of inhibiting carbapenemases (avibactam) to broaden the antibacterial spectrum and potency. Following its approval in March 2019 for children older than 3 months, it has been reported as a successful treatment of invasive CRE infections as well as complicated intra-abdominal infections, urinary tract infections (UTI), hospital-acquired pneumonia and ventilator-associated pneumonia (VAP) [13,14,15]. Many clinicians use CAZ-AVI to treat suspected CRE bloodstream infections (BSIs) and infections due to aerobic Gram-negative bacteria with limited treatment options [16,17]. Findings suggest that adult patients who received CAZ-AVI compared with other regimens against CRE infections had significantly lower mortality, good tolerance and less nephrotoxicity [18,19,20,21]; nevertheless, few available pediatric case series are reported.

This study aims to analyze the epidemiological, clinical and microbiological characteristics of the patients who received CAZ-AVI during their admission to a PICU, as well as the outcomes.

## 2. Results

### 2.1. Patients and Clinical Characteristics

The study included ten patients with 12 episodes of clinical suspicion or confirmed MDR bacterial infections treated with CAZ-AVI. The median age was 7 years (IQR 0.3–17), and 60% were female. The origins of 60% of the patients were from foreign countries (Peru, Nicaragua, India, United Arab Emirates and Poland). The most frequent referring service was the pediatric hospitalization ward (80%). Admission was due to medical conditions (80%) and elective surgery (20%). All of the patients had previous hospital admissions, and they also had previous comorbidities, divided into five groups: congenital cardiopathies (30%), chronic renal failure (10%), solid neoplasia (20%), hematological neoplasia (30%) and neuromuscular pathology (10%). CAZ-AVI prescription indications were surgical prophylaxis (1), sepsis suspicion (6), catheter-associated BSI (1), secondary bacteremia (1), community-acquired pneumonia (1), ventilator-associated pneumonia (1) and surgical wound infection (1). Table 1 includes a general epidemiological description of the sample.

### 2.2. Empirical Treatment

Eight patients received empirical treatment based on previous colonizations with MDR bacteria, as shown in Table 2.

Regarding the episodes, 9 of 12 (75%) were treated empirically.

In one episode, a patient diagnosed with hematological neoplasia and suspected central-line-associated bloodstream infection (CLABSI) received CAZ-AVI as an empirical treatment due to the severity of the clinical presentation and previous exposure to broad-spectrum antibiotics, although no MDR bacteria had been isolated in previous cultures.

In four episodes, patients with previous ESBL producinf-enterobacteria colonization.received empirical treatment with CAZ-AVI due to meropenem not being indicated because of a breakthrough infection during carbapenem treatment.

In four episodes, patients received CAZ-AVI plus aztreonam based on previous MDR bacteria MBL-producers: New Delhi MBLs, KPC-*Klebsiella aerogenes and new Delhi* MBLs.

### 2.3. Targeted Treatment

Two patients (20%) received targeted therapy because they were diagnosed with an MDR bacterial infection. We found the isolation of MDR *Pseudomonas aeruginosa* from bronchoalveolar lavage in a patient diagnosed with VAP (Table 2, episode 6). The antibiotic susceptibility pattern was piperacillin–tazobactam, quinolone and carbapenem resistance with colistin, amikacin and CAZ-AVI sensitivity. The second patient with suspected surgical wound infection received CAZ-AVI plus aztreonam due to the isolation of *Escherichia hermannii* carbapenemase-type Verona integron-encoded metallo-β-lactamase (VIM) in a surgical wound culture. In this case, the antibiotic susceptibility pattern was gentamicin, trimetropim, penicillin, and cephalosporin resistance except for CAZ-AVI, cefiderocol and monobactams such as aztreonam. Control cultures after antibiotic initiation were negative for MDR bacteria.

### 2.4. Other Characteristics of the Treatment

The mean duration of the treatment was 3 days (IQR 1–7 days). In the case of the two patients that received targeted treatment, the duration was 7 days for the patient diagnosed with a VAP and 4 days for the patient diagnosed with a wound infection, and the treatment was changed according to the antibiogram to complete 6 weeks of treatment.

In the twelve episodes (100%), CAZ-AVI was administered as a combination therapy with other antibiotics (one or two): in seven episodes (58%) it was administered in combination with aztreonam (plus empirical vancomycin in three of them); in one episode (8%), in combination with colistin (plus empirical vancomycin); in three episodes (25%), as an empirical combination therapy that included CAZ-AVI plus vancomycin or teicoplanin; and in one episode (8%), CAZ-AVI plus vancomycin and clindamycin.

In ten episodes (80%) antibiotic was stopped or de-escalated when CRE were not identified in clinical sample. The antibiotics with a narrower spectrum used were piperacillin–tazobactam, amoxicillin–clavulanic, cefotaxime and meropenem.

No bacteria resistant to CAZ-AVI were isolated. No side effects or fatal outcomes were reported related to CAZ-AVI administration. The 28-day mortality rate was 0%.

The detailed results regarding the microbiological characteristics of the infection episodes are included in Table 2.

## 3. Discussion

When a critically ill pediatric patient is suspected of having sepsis or severe bacterial infection, it is important to start early suitable empirical antibiotic treatment due to the increased number of MDR bacterial infections. CAZ-AVI has been demonstrated as an effective and safe antibiotic for patients with CRE infections in well-controlled phase III studies in adults and phase II studies in children and young infants [14,22,23,24,25]. Furthermore, recent studies have shown that the combination of CAZ-AVI plus aztreonam appears to be a promising option against CRE infections including MBL-producing bacteria, especially Enterobacterales [26].

In recent years, there has been a rise in the number of medically complex children leading to increased use of invasive medical devices, immunosuppressive treatments and long-term admissions, including prolonged PICU hospitalizations [5]. Our study found that all patients had comorbidities and indwelling devices, with previous hospitalizations being a risk factor for developing MDR infections [4,8,19]. The origins of 60% of the patients were from foreign countries (Peru, Nicaragua, India, United Arab Emirates, and Poland) with different prevalences of MDR bacteria in populations with high rates of CRE. In South America and Poland, KPC is the most prevalent carbapenemase; India and the United Arab Emirates are MBL-endemic regions [27,28,29]. This could explain the type of previous colonizations detected in these patients. Specific empirical antimicrobial treatment may be needed for newly admitted patients from these countries who present acute infections [8,30].

There are currently no available data on the percentage of pediatric patients colonized with MDR bacteria who received CAZ-AVI during an acute infection. In our case, 80% of patients had previous MDR bacterial colonization, but only two of them had an acute MDR bacterial infection confirmed by positive cultures. One of them was an MDR *Pseudomonas aeruginosa* VAP. Clinical trials have shown that CAZ-AVI has greater in vitro activity against *Pseudomonas aeruginosa* and less resistance compared to other anti-pseudomonal agents, such as colistin and quinolones [12]. The second acute infection, from a non-previously colonized Spanish patient with risk factors such as surgery and prolonged hospitalization, was caused by *Escherichia hermannii* carbapenemase type VIM. Although there are few epidemiological studies on Spanish children due to low strain circulation, there is a predominance of resistance MBL type VIM [31,32].

The most frequent use of CAZ-AVI in our PICU was the empirical treatment for patients with risk factors for developing an MDR bacterial infection, especially those with previous colonizations and clinical suspicion of bacteremia or sepsis. Although CAZ-AVI is only approved for urinary tract, abdominal and lower respiratory tract MDR bacterial infections, a systematic review comparing CAZ-AVI with other regimens in CRE bacteremia showed significantly lower 30-day mortality, suggesting possible use as a first-line treatment in patients with suspected CRE bacteremia until cultures turn negative. Further studies are needed to provide recommendations on the treatment of BSIs caused by CRE similar to UTI recommendations [17]. Tumbarello et al. [18] reported a cohort of KPC-producing *Klebsiella pneumoniae* infections in adults, with 67.8% BSIs. These patients were treated with CAZ-AVI, and no significant differences were found in terms of side effects or mortality. Moreover, CAZ-AVI was associated with better survival rates in patients with bacteremia who required rescue treatment for infections caused by KPC-producing *Enterobacteriaceae* [7].

The average duration of administered CAZ-AVI was three days, as these patients mainly received empirical treatment because of previous colonizations until cultures turned negative or a susceptible microorganism was isolated. Empirical treatment recommendations should be based on the organisms identified in the previous six months and guided by illness severity and the likely source of the infection [10]. Regarding the two acute MDR bacterial infections, the maximum duration was 7 days. Few data provide recommendations on the duration of CAZ-AVI, but recommendations on the duration of therapy in acute infections should not differ from infections caused by more susceptible phenotypes. Host factors should be considered to determine the duration of the treatment. 

Regarding CAZ-AVI-resistant bacterial cultures, many reports emphasize the importance of a short duration of treatment because of the potential of CAZ-AVI to select for bacterial resistance. Studies in adult patients with CRE infections reported the isolation of CAZ-AVI-resistant strains in patients who received treatment for at least 10 days [18,33]. In our case, no CAZ-AVI-resistant bacteria were found, probably due to the short duration of treatment, with a maximum of seven days. It is recommended to monitor for the emergence of new bacterial resistance while the patient is receiving treatment [7].

In our sample, no fatal outcomes were observed. The two patients with confirmed infections with CRE, who received targeted treatment, presented favorable clinical evolution with negative control cultures. This result could be explained due to the adequate therapeutic coverage with the antibiotic combination. There are limited data about the outcomes in pediatric critical patients treated with CAZ-AVI. Still, phase II studies have shown safety for the treatment of severe CRE infections with off-label use [19]. In addition, CAZ-AVI has been reported as a successful treatment in MDR septic shock in critically ill patients, including liver transplantation patients [21,34]. Studies involving adult patients [18] report an all-cause-mortality rate 30 days after infection onset of 25%, related to an older age and comorbidities. Nevertheless, comparative effectiveness studies with CAZ-AVI for CRE infection have demonstrated improved outcomes with significantly lower 30-day mortality and higher clinical cure rates than control groups with other regimens [7].

Moreover, no side effects were found in our study. Other adult and pediatric studies [18,19,21] described adverse reactions such as skin, abdominal and dyselectrolytemia symptoms. The risk of adverse events includes the establishment of renal impairment with the recommendation to monitor creatinine clearance at least daily in pediatric patients with changing renal function, to adjust the dose.

In our study, 100% of the infections were treated with combination therapy. A systematic review [35] about CAZ-AVI combination therapy compared to CAZ-AVI monotherapy in patients with CRE infections (mostly KPC) found no difference in mortality rate, concluding that this finding could be useful for optimizing the antibiotic treatment, with the potential to reduce the use of combination treatments. Nevertheless, local epidemiology should be considered when deciding to use monotherapy or combination therapy. In our case, combination therapy was used to cover microorganisms other than CRE or to increase the bactericidal effect against CRE in the case of severe infections. Many studies found potential advantages in vitro when combining CAZ-AVI with colistin, rifampicin or fosfomycin against *Pseudomonas* spp., and a synergistic activity was also observed with carbapenems against *Klebsiella pneumoniae* KPC and *Serratia marcescens* KPC [19,36,37].

The results showed that 60% of the episodes were treated with aztreonam combination therapy due to the determination of Ambler class B in previous colonizations of those patients. It is known that avibactam is a β-lactamase inhibitor that binds reversibly to serine-β-lactamases with activity against most KPC and OXA-48-like carbapenemases but remains inactive against MBL-producers [38]. In vitro studies suggest a synergistic effect of aztreonam and CAZ-AVI for severe MDR bacterial infections with few therapeutic regimens available, such as MBL class B members including the New Delhi metallo-β-lactamase, VIM and imipenemase, and class D enzymes [39,40]. Regarding the 12th episode, the patient with an acute surgical wound infection of VIM producing *Escherichia hermannii*, the combination of CAZ-AVI with aztreonam probably contributed to the resolution of the infection.

Our study had several limitations, namely the small size of the sample, due to the few episodes of MDR bacterial infections registered in our PICU that received CAZ-AVI during the year 2022. In this case, the study could be subject to confounding factors but may show a trend in the use of this antibiotic. Due to 60% of the episodes receiving aztreonam, it is challenging to interpret the efficacy of CAZ-AVI. The combination of CAZ-AVI plus aztreonam could be considered a better rescue therapy for MBL infections [41]. No consensus susceptibility testing method for this triple combination has yet to be recommended. In this context, in March of 2024, a new combination of avibactam plus aztreonam was approved in Europe. The indications in adult patients were for CRE intra-abdominal, urinary tract and HAP infections with limited options for treatment, especially for infections with MBL-producing bacteria [42]. Efficacy in pediatric patients has not yet been studied.

More future studies with greater samples are needed to assess the safety profile and effectiveness of CAZ-AVI in critically ill pediatric patients.

## 4. Materials and Methods

This was an observational, prospective study conducted in a PICU of a tertiary University Pediatric Hospital (Barcelona, Spain), for a period of one year, from January to December 2022. Inclusion criteria were children under 18 years old treated with CAZ-AVI in the PICU during the study period. Criteria exclusion were patients whose parents did not sign informed consent for this study.

The antibiotic practices in our PICU are based on the use of a beta-lactam antibiotic with anti-pseudomonic action such as piperacillin–tazobactam plus an antibiotic with coverage for Gram-positive bacteria such as vancomycin. In severe infections with suspected MDR bacteria involved, meropenem would be indicated. The use of CAZ-AVI in our hospital was reserved for seriously ill patients admitted to the ICU in whom previous therapies cannot be used because of resistance or a relation to recent previous treatment with meropenem. The decision to initiate CAZ-AVI was proposed and revised by the PICU pediatric consultant team with infectious diseases consultants.

All the patients more than 40 kg (6–18 years) received an intravenous dose of 2 g of ceftazidime and 0.5 g of avibactam every 8 h. Patients under 40 kg (≥3 months–6 years) received 50 mg/kg of ceftazidime and 12.5 mg/kg of avibactam every 8 h daily. The time of the infusion was 120 min.

The parameters collected for this analysis included the following: epidemiological data (sex, age, clinical characteristics, referring service, previous admissions), microbiological variables (previous colonization, type of infection, current infection cultures, type of resistance), treatment (duration, combined therapy) and outcomes (secondary effects and death at 28 days).

### 4.1. Definitions

-Sepsis was considered following the criteria of international guidelines on sepsis in children [43,44]. Bacteremia was defined by the growth of a known bacterial pathogen in the corresponding blood sample.-Community-acquired pneumonia (CAP) was characterized in accordance with the British Thoracic Society guidelines [45,46,47,48,49]. Bacterial pneumonia should be considered in children when there is persistent fever >38.5 °C, with chest recession and raised respiratory rate. The presence of an alveolar infiltration in the CXR and the elevation of acute phase reactants are thought to be secondary to bacterial cause. CAP diagnosis should be considered when patients were outside the hospital environment or within 48 h of admission.-Ventilator-associated pneumonia (VAP) was suspected when there was an acute infection of the pulmonary parenchyma, associated with clinical signs and symptoms, and increased oxygen requirements, in a patient receiving mechanical ventilation for more than 48 h [50]. It was diagnosed based on the CDC’s definition [51].-Surgical wound infection was suspected when there was an infection that occurred after surgery in the body region where the surgery took place. Symptoms included redness and pain around the surgical area, fever or drainage of cloudy fluid from the surgical wound. The diagnosis was based on the CDC’s definition [52].-Multidrug-resistant bacteria: Isolates with resistance to at least 1 antibiotic in more than 3 categories, based on the Magiorakos classification modified considering the current EUCAST susceptibility definitions [34,53].-Previous MDR colonization: Colonization with MDR bacteria, which were methicillin-resistant *Staphylococcus aureus* (MRSA), extended-spectrum beta-lactamase-producing Enterobacterales (ESBL-E), carbapenem-resistant Enterobacterales (CRE) and MDR *P. aeruginosa* [53].-Current infection cultures were defined as cultures obtained from blood, urine, respiratory, skin or other tissue samples removed from the suspicious infected tissue and cultured before the initiation of antibiotic therapy.-Control cultures were defined as cultures collected after 24 h of antibiotic therapy from the suspicious infected tissue.

### 4.2. Statistical Analysis

Descriptive statistical analysis of the data was performed. Frequencies and percentages were used for qualitative variables.

## 5. Conclusions

The use of CAZ-AVI in our PICU is principally reserved for short-duration empirical treatment for pediatric patients with previous colonization with MDR bacteria and clinical suspicion of bacteremia or sepsis, and for the targeted treatment of MDR bacterial infections. No side effects or fatal outcomes were found. Furthermore, the combination of CAZ-AVI plus aztreonam could be more effective for CRE infections, especially type Ambler class B as MBL strains. This bitherapy needs to be reserved for patients with limited treatment options.

The treatment of CRE infections in children is complex, and it is necessary to have a multidisciplinary approach including specialists in infectious diseases and microbiology, with a close follow-up of the patients, to use this antibiotic in critically ill children. Further studies are needed to optimize targeted treatment in suspected severe pediatric infections caused by CRE.

## Figures and Tables

**Table 1 antibiotics-13-01037-t001:** Epidemiological data of the sample.

Episode No.	Sex	Age (Years)	Comorbidities	Origin Country
1	Female	0.6	Congenital cardiopathy	Spain
2	Female	7	Chronic renal failure	Poland
3	Male	0.6	Congenital cardiopathy	Spain
4	Female	6	Solid neoplasia	India
5–6	Male	1.5	Solid neoplasia	United Arab Emirates
7	Male	0.3	Hematological neoplasia	Spain
8–9	Male	8	Hematological neoplasia	Peru
10	Female	0.8	Congenital cardiopathy	Nicaragua
11	Female	5	Hematological neoplasia	Peru
12	Female	17	Neuromuscular pathology	Spain

**Table 2 antibiotics-13-01037-t002:** Microbiological characteristics of infection episodes.

Episode	Suspected Infection	Use of CAZ-AVI	Previous MDR Colonization	Current Infection Microbiological Identification	Control Cultures	Additional Treatment
**1**	Secondary bacteremia	Empirical	*Klebsiella pneumoniae* ESBL and porin alterations	*Klebsiella pneumoniae* susceptible isolated in bronchoalveolar lavage	Not done	Vancomycin
**2**	Surgical prophylaxis	Prophylaxis	KPC-*Klebsiella aerogenes* and New Delhi MBL	Negative	Negative	Aztreonam
**3**	Sepsis	Empirical	ESBLs *Klebsiella pneumoniae*	Negative	Negative	Vancomycin
**4**	CAP	Empirical	*Escherichia coli* ESBLs and New Delhi MBL	Negative	Negative	Aztreonam
**5**	Sepsis	Empirical	*Escherichia coli* ESBL	Negative	Negative	Clindamycin, vancomycin
**6**	VAP	Targeted treatment	MDR *Pseudomonas aeruginosa*	MDR *Pseudomonas aeruginosa* isolated in bronchoalveolar lavage	Negative	Colistin
**7**	Catheter-associated bacteriemia	Empirical for gravity	Negative cultures	Negative	Negative	Aztreonam
**8**	Sepsis	Empirical	*Salmonella enterica* ESBL *Escherichia coli* New Delhi MBL	*Stenotrophomonas maltophilia* isolated in blood culture	Not done	Aztreonam
**9**	Sepsis	Empirical	*Salmonella enterica* ESBL *Escherichia coli* New Delhi MBL	*Pseudomonas aeruginosa* susceptible isolated in blood culture	Negative	Aztreonam, vancomycin
**10**	Sepsis	Empirical	*Escherichia coli* New Delhi MBL	Negative	Not done	Aztreonam, vancomycin
**11**	Sepsis	Empirical	*Salmonella cholerasuis* ESBL	Negative	Not done	Teicoplanin
**12**	Surgical wound infection	Targeted treatment	Negative cultures	*Escherichia hermannii* carbapenemase type VIM isolated in surgical wound culture	Negative	Aztreonam, vancomycin, teicoplanin

## Data Availability

Data are contained within the article.

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
