# Peer review of "The Use of Ceftazidime–Avibactam in a Pediatric Intensive Care Unit—An Observational Prospective Study"

_antibiotics, 2024, doi:10.3390/antibiotics13111037_

Round 1
Reviewer 1 Report
Comments and Suggestions for Authors
Few points which need to be addressed are:
1. 2Methods: Please mention the status of antibiotic stewardship practices, if any, being implemented in the study setting especially with regard to the prescription of “reserve” group antibiotics.
2. In 80% of the cases (8 out of 10), the use of ceftazidime-avibactam was empirical. The use of Ceftazidime-avibactam needs to be very judiciously and meticulously done, considering the fact that the drug is being classified as a “Reserve” category drug under the WHO AWaRe classification. I have no reservations on its use for the clinical scenarios being mentioned in the study. However, I would suggest the authors to add adequate justification/s for the use in cases where it was driven on the basis of prior history of MDR organisms though there was no isolation of MDR in the current infection. What are the AMR patterns in the study hospital such as the prevalence of MDR organisms’ isolation? Was the decision to start the drug governed by the hospital antibiotic policy? If yes, please add a brief account of the same. Were any infectious disease consultants involved in the process?
3. For the two cases with targeted treatment, please also add details pertaining to the antibiotic susceptibility patterns in addition to the type of organisms.
Comments on the Quality of English LanguageModerate editing needed.
Author Response
First of all thank you for the revision.
1- In reference for the first comment of methods: Please mention the status of antibiotic stewardship practices, if any, being implemented in the study setting especially with regard to the prescription of “reserve” group antibiotics.
We have accordingly revised and modified to clarify the concepts. This change can be found in page 2, line 86:
The antibiotic practices in our PICU are based on the use of a beta-lactam antibiotic with anti-pseudomonic action such as piperacillin-tazobactam plus an antibiotic with coverage for gram positives such as vancomycin. In severe infections with suspected MDR bacteria involved, meropenem would be indicated. The use of CAZ-AVI in our hospital was reserved for seriously ill patients admitted to the ICU in whom previous therapies cannot be used because of being resistant or related to recent previous treatment with meropenem. The decision to initiate CAZ-AVI was proposed and revised by the PICU pediatric consultant team with infectious diseases consultants
Regarding comment 2: In 80% of the cases (8 out of 10), the use of ceftazidime-avibactam was empirical. The use of Ceftazidime-avibactam needs to be very judiciously and meticulously done, considering the fact that the drug is being classified as a “Reserve” category drug under the WHO AWaRe classification. I have no reservations on its use for the clinical scenarios being mentioned in the study. However, I would suggest the authors to add adequate justification/s for the use in cases where it was driven on the basis of prior history of MDR organisms though there was no isolation of MDR in the current infection. What are the AMR patterns in the study hospital such as the prevalence of MDR organisms’ isolation? Was the decision to start the drug governed by the hospital antibiotic policy? If yes, please add a brief account of the same. Were any infectious disease consultants involved in the process?
The two last questions are mencioned in methods. To clarify the use of empirical treatment of CAZ-AVI, we have revised and modified this section (page 4, line 160).
Empirical treatment: Eight patients received empirical treatment based on previous colonizations with MDR bacteria as is shown in table 2. Regarding the episodes, 9 of 12 (75%) were treated empirically. In one episode, a patient diagnosed with haematological neoplasia and suspected central line associated bloodstream infection (CLABSI), received CAZ-AVI as an empirical treatment due to the severity state although no MDR bacteria had been isolated in previous cultures. In 4 episodes, patients who had previously been colonized with ESBL-producing enterobacteria, received empirical treatment with CAZ-AVI due to meropenem was not indicated because of a breakthrough infection during carbapenem treatment. In 4 episodes, patients received CAZ-AVI plus aztreonam based on previous MDR bacteria MBL producers: New Delhi metallo-β-lactamases, Klebsiella aerogenes KPC and new Delhi metallo-β-lactamases.
In response to the comment 3: For the two cases with targeted treatment, please also add details pertaining to the antibiotic susceptibility patterns in addition to the type of organisms.
We have added this information in line 177, page 5.
Targeted treatment:
Two patients (20%) received targeted therapy because they were diagnosed with an MDR bacterial infection. We found the isolation of MDR Pseudomonas aeruginosa from bronchoalveolar lavage in a patient diagnosed of VAP (table 2, episode 6). The antibiotic susceptibility pattern was piperacillin-tazobactam, quinolones, and carbapenems resistance with colistin, amikacin and CAZ-AVI sensitivity. The second patient with suspected surgical wound infection, received CAZ-AVI plus aztreonam due to the isolation of Escherichia hermannii carbapenemase type Verona integron-encoded metallo-β-lactamase (VIM) in a surgical wound culture. In this case, the antibiotic susceptibility pattern was gentamicin, trimetropim, penicillin, and cephalosporin resistance except for CAZ-AVI, cefiderocol and monobactams as aztreonam. Control cultures after antibiotic initiation were negative for MDR bacteria.
We have tried to clarify all this concepts. If there is any doubt, let us know again to answer and modify.
Reviewer 2 Report
Comments and Suggestions for Authors
This study analyses the epidemiological, clinical and microbiological characteristics of patients who received CAZ-AVI during their admission to a PICU.
I find the manuscript very interesting, despite the limited number of cases examined and the inclusion of several episodes of metallo-betalactamase (MBL)-producing colonisers in which the use of ceft/AVI seems questionable. But it is precisely at this point that the manuscript becomes attractive. The majority of the documented episodes are treated empirically by combining CAZ/AVI with aztreonam (stable against MBL), therefore the resolution of MBL-producing colonisers episodes must be supported by this combination. Given the recent commercialisation of aztreonam/avibactam, this point should be the focus of the paper. It is not adequately addressed by the authors in the current format. I suggest a complete reformulation of the manuscript around this aspect.
In my opinion, this confounding factor is worse than the small number of cases examined (line 302) and its clarification would greatly help in understanding the resolution of episodes with CAZ/AVI administration.
Specific comments:
Line 46. Please, include MBLs (Ambler clase B)
Line 61. Note that AVI is the inhibitor, not CAZ/AVI. Check and correct.
Line 72: The aim of the study includes side effects, which appear for the first time in line 178 and in the discussion. However, the authors do not provide objective data to support their inclusion.
Line 99. Please indicate the time from hospital admission to establishing a CAP.
Line 133.Please remove patient numbering from Table 1, as it creates confusion with the number of episodes.
Line 134: Empirical treatment. The description is unclear. The percentage calculation should be revised. Please; Explain number of patients who received empirical treatment. Then explain the number of episodes treated empirically. I suggest moving the description of colonisers to Table 2 with the corresponding episode. In this section 3.2. it would be useful to group colonisers (%) into those CRE that are CAZ/AVI targets and those that are not (MBL producers).
Line 153: See comment for line 134
Line 157: “Control cultures after antibiotic initiation were negative”. It is difficult to understand without a prior description of the term. Please, explain the concept in the material and methods section. Does the phrase refer to when cultures were negative for MDR bacteria and/or a susceptible bacterium?? (see line 173 and the current infection culture column in table 2).
Line 167-172. Replace the term synergy with the term combination (for example).
Line 178: “No bacteria resistant to CAZ-AVI was isolated” When and how was this done? As in the previous comment, please describe adequately in the material and methods section.
Line 182: Table 2. Previous MDR colonization column. Please remove the numbering and include detailed text in the footer of the table. Insert column explaining additional treatment to CAZ/AVI per episode. Explain the concept current infection culture in Materials and Methods section.
Line 190: Discussion. This section should focus on the role of the CAZ/AVI and Aztreonam combination.
Line 245: “Regarding CAZ-AVI resistant bacterial cultures, many reports highlight the importance of short duration of therapy due to the potential of CAZ-AVI to develop bacterial…. Please rephrase the text; it appears that resistance refers to this work. Please note that antimicrobials do not cause resistance, they only select for it (check and correct). Also, the text on lines 249-251 is confusing. It is proper coverage that prevents selection for resistance.
Lines 253-254. “The two patients with active CRE infections, who received targeted treatment, presented favorable clinical evolution with negative control cultures”. Please check the meaning of negative control cultures here. I assume that this does not refer to the content of table 2 (column current infection culture). Hence the importance of defining the terms properly in the material and methods section.
Line 256: Rather than age factor, I understand that it is due to adequate therapeutic coverage (antibiotic combination). Check and correct.
Lines 265-278. Please, summarise
Line 280. This text is interesting. Make appropriate reference to the resolution of the episodes and explain the role of CAZ/AVI and combination antimicrobials (aztreonam) in the resolution of the case.
Line 306. Conclusions. The relationship between CAZ/AVI and aztreonam needs to be highlighted. Line 313; please, include the prospects for use of the aztreonam/avibactam combination.
Author Response
First of all, thank you for the review. In reference of the combination of CAZ-AVI and aztreonam treatment, we have tried to emphasize this concept to clarify and understand the resolution of all the episodes.
For the specific comments:
Line 46. Please, include MBLs (Ambler clase B): We have modified, now is mentioned in line 49.
Line 61. Note that AVI is the inhibitor, not CAZ/AVI. Check and correct: Agree, we have modified, line 62.
Line 72: The aim of the study includes side effects, which appear for the first time in line 178 and in the discussion. However, the authors do not provide objective data to support their inclusion. We know that one of our limitations is the size of the sample, we have removed side effects (we have included in outcomes of CAZ-AVI).
Line 99. Please indicate the time from hospital admission to establishing a CAP. Modified in line 113.
Line 133.Please remove patient numbering from Table 1, as it creates confusion with the number of episodes. We have removed it.
Line 134: Empirical treatment. The description is unclear. The percentage calculation should be revised. Please; Explain number of patients who received empirical treatment. Then explain the number of episodes treated empirically. I suggest moving the description of colonisers to Table 2 with the corresponding episode. In this section 3.2. it would be useful to group colonisers (%) into those CRE that are CAZ/AVI targets and those that are not (MBL producers). Line 153: See comment for line 134
We have clarified all this section (line 160). We also have modified table 2.
Line 157: “Control cultures after antibiotic initiation were negative”. It is difficult to understand without a prior description of the term. Please, explain the concept in the material and methods section. Does the phrase refer to when cultures were negative for MDR bacteria and/or a susceptible bacterium?? (see line 173 and the current infection culture column in table 2). We have added in methods section.
Line 167-172. Replace the term synergy with the term combination (for example). Done
Line 178: “No bacteria resistant to CAZ-AVI was isolated” When and how was this done? As in the previous comment, please describe adequately in the material and methods section. Included in methods.
Line 182: Table 2. Previous MDR colonization column. Please remove the numbering and include detailed text in the footer of the table. Insert column explaining additional treatment to CAZ/AVI per episode. Explain the concept current infection culture in Materials and Methods section. We have included in methods and changed the table 2.
Line 190: Discussion. This section should focus on the role of the CAZ/AVI and Aztreonam combination. Line 221.
Line 245: “Regarding CAZ-AVI resistant bacterial cultures, many reports highlight the importance of short duration of therapy due to the potential of CAZ-AVI to develop bacterial…. Please rephrase the text; it appears that resistance refers to this work. Please note that antimicrobials do not cause resistance, they only select for it (check and correct). Also, the text on lines 249-251 is confusing. It is proper coverage that prevents selection for resistance. We have rephrased it, hope that now is better clarified.
Lines 253-254. “The two patients with active CRE infections, who received targeted treatment, presented favorable clinical evolution with negative control cultures”. Please check the meaning of negative control cultures here. I assume that this does not refer to the content of table 2 (column current infection culture). Hence the importance of defining the terms properly in the material and methods section. Checked.
Line 256: Rather than age factor, I understand that it is due to adequate therapeutic coverage (antibiotic combination). Correct.
Lines 265-278. We have summarised.
Line 280. This text is interesting. Make appropriate reference to the resolution of the episodes and explain the role of CAZ/AVI and combination antimicrobials (aztreonam) in the resolution of the case. Line 319 and 328.
Line 306. Conclusions. The relationship between CAZ/AVI and aztreonam needs to be highlighted. Line 313; please, include the prospects for use of the aztreonam/avibactam combination. Line 344
Round 2
Reviewer 1 Report
Comments and Suggestions for Authors
The authors have modified the manuscript as desired. No further comments from my side.
Comments on the Quality of English LanguageMinor to moderate editing in English language required.
Reviewer 2 Report
Comments and Suggestions for Authors
Congratulations on this excellent work. I encourage you to continue this line of research as soon as the new axteronam avibactam formulation is available.